# SK-03-92 Drug Kills Intracellular *Mycobacterium tuberculosis*

**DOI:** 10.3390/antibiotics12091385

**Published:** 2023-08-30

**Authors:** William R. Schwan

**Affiliations:** Department of Microbiology, University of Wisconsin-La Crosse, 1725 State St., La Crosse, WI 54601, USA; wschwan@uwlax.edu; Tel.: +1-(608)-785-6980

**Keywords:** SK-03-92, *Mycobacterium tuberculosis*, stilbene, macrophage

## Abstract

Background: Tuberculosis affects millions of people worldwide. The emergence of drug-resistant *Mycobacterium tuberculosis* strains has made treatment more difficult. A drug discovery project initiated to screen natural products identified a lead stilbene compound, and structure function analysis of hundreds of analogs led to the discovery of the SK-03-92 stilbene lead compound with activity against several non-tuberculoid mycobacteria. Methods: For this study, an MIC analysis and intracellular killing assay were performed to test SK-03-92 against *M. tuberculosis* grown in vitro as well as within murine macrophage cells. Results: The MIC analysis showed that SK-03-92 had activity against *M. tuberculosis* in the range of 0.39 to 6.25 μg/mL, including activity against single-drug-resistant strains. Further, an intracellular kill assay demonstrated that the SK-03-92 lead compound killed *M. tuberculosis* cells within murine macrophage cells. Conclusion: Together, the data show the SK-03-92 lead compound can kill *M. tuberculosis* bacteria within mammalian macrophages.

## 1. Introduction

*Mycobacterium tuberculosis* causes the respiratory disease tuberculosis (TB), which led to 10.6 million new cases and killed 1.6 million people worldwide in 2021 [1]. Latent tuberculosis infections in the United States are thought to comprise 5% of the population, whereas, around the world, it is estimated that 15.9% of the population or around 1.27 billion people are latently infected [2]. In 2014, the WHO formulated the End TB Strategy to halt the global TB epidemic by 2035 [3]. Despite this initiative, there has been a mere 11% decline in TB cases since 2015 [1].

One factor contributing to the marginal decline in TB cases is drug-resistant strains of *M. tuberculosis*. Currently, a four-drug regimen is recommended in developed countries consisting of rifampin (RIF), isoniazid (INH), ethambutol, and pyrazinamide [4]. However, multidrug-resistant *M*. *tuberculosis* (MDR-TB) strains that are resistant to two of the frontline drugs, INH and RIF, emerged in the 1980s and represent 3–5% of all new TB infections [5,6]. Further, extensively drug-resistant *M. tuberculosis* (XDR-TB) strains arose in 2006 that were resistant to all frontline anti-TB drugs plus fluoroquinolones and one injectable anti-TB drug and are thought to comprise 10% of the MDR-TB strains currently [6,7]. Thus, new drugs with novel mechanisms of action are needed to treat TB cases.

To identify new anti-mycobacteria drugs from natural products, plant and mushroom extracts were screened, and a (E)-3-hydroxy-5-methoxystilbene compound was isolated from the sweet fern plant [8]. Several hundred analogs of the (E)-3-hydroxy-5-methoxystilbene drug led to the discovery of the SK-03-92 analog with a stilbene scaffold that had activity against several non-tuberculoid mycobacteria species including *Mycobacterium kansasii* and *Mycobacterium avium* [9]. In this study, the SK-03-92 lead compound was tested by MIC against four *M. tuberculosis* strains that included single-drug-resistant strains and found to have some activity against all of the strains. Further, the SK-03-92 lead compound was able to kill *M. tuberculosis* cells within murine macrophages. Thus, SK-03-92 shows some promise as an anti–*M. tuberculosis* drug.

## 2. Results

### 2.1. SK-03-92 Has Activity against Drug-Resistant M. tuberculosis Strains

Previously, we demonstrated that SK-03-92 had activity against non-tuberculoid mycobacterial species [9]. For this study, an MIC analysis was conducted to assess the efficacy of SK-03-92 against *M. tuberculosis*, including drug-resistant strains. The results showed that SK-03-92 had MIC values of 6.25 μg/mL against the *M. tuberculosis* strains H37Rv and SRI 1369 (INH^R^, Table 1). Against *M. tuberculosis* strains that were rifampin resistant or moxifloxacin resistant, SK-03-92 had MIC values of 0.39 and 0.78 μg/mL, respectively. These results show that SK-03-92 has activity against *M. tuberculosis* strains that include drug-resistant variants.

### 2.2. SK-03-92 Kills M. tuberculosis within Mammalian Macrophages

The MIC analysis above showed that SK-03-92 had activity against several strains of *M. tuberculosis*. Since *M. tuberculosis* survives inside mammalian macrophages as part of its lifestyle, testing was carried out to determine whether SK-03-92 could kill *M. tuberculosis* within macrophages.

First, an MTT cytotoxicity assay was performed to assess the toxicity of SK-03-92 against murine macrophage cells. The results demonstrated that SK-03-92 had minimal effects on J774A.1 macrophages at low or medium doses (Table 2). However, cell viability dropped to <10% with a high dose of the SK-03-92 drug. Rifampin had minimal cytotoxic effects on the macrophages using drug concentrations up to 10 μg/mL.

Next, the killing effect of SK-03-92 on *M. tuberculosis* was assessed by infecting J774A.1 macrophages and then treating with SK-03-92 at the same low, medium, and high doses used for the cytotoxicity assay. SK-03-92 caused significant reductions in *M. tuberculosis* counts at low [0.5 μg/mL, 2.26 log reduction (*p* < 0.004)], medium [5 μg/mL, 2.23 log reduction (*p* < 0.006)], and high concentrations [50 μg/mL, 4.37 log reduction (*p* < 0.005)] (Figure 1A) compared to rifampin (Figure 1B). The PBS control displayed a viable count drop that was similar to rifampin used at a low concentration.

## 3. Discussion

Tuberculosis remains a significant public health concern across the world. Drug-resistant strains of *M. tuberculosis* complicate the treatment options available to patients. In this study, SK-03-92 was shown to have in vitro efficacy against several *M. tuberculosis* strains, including strains resistant to frontline anti-tuberculosis drugs.

Because of drug-resistant strains of *M. tuberculosis*, new anti-mycobacterial drugs are needed. Newer anti-mycobacterial drugs that have been approved include bedaquiline, which targets mycobacterial ATPase [10,11], and delamanid, which targets mycolic acid synthesis [12,13]. Moxifloxacin, a fluoroquinolone drug that targets DNA gyrase, has been repurposed for treating MDR-TB. Currently, 21 other drugs are moving through human clinical trials to assess their safety and efficacy [14]. 

Our drug discovery program identified the lead compound SK-03-92 that had broad activity against Gram-positive bacteria as well as non-tuberculoid mycobacterial species [9]. The SK-03-92 lead compound was derived from a natural product and possesses a stilbene backbone [8,9]. Other stilbene-based chemical structures have been investigated for their anti-mycobacterial activities. Resveratrol has been shown to inhibit the growth of *M. tuberculosis* by activating the host protein sirtuin 1 and inhibiting apoptosis in macrophages [15,16,17,18]. Tamoxifen (a phenylstilbene compound) is being explored as an anti-mycobacterial drug, acting as a host-directed therapeutic that assists macrophages in killing mycobacterial cells [19,20,21]. Analogs of 3,5-dimethoxystilbene had MIC values that ranged from 61.68 and 74.04 μg/mL against *M. intracellulare* [22]. The 3-fluoro-Z-stilbene exhibited an MIC of 26 μg/mL against *M. bovis* [23]. Several aza-stilbene derivatives displayed MIC values of 15.6 μg/mL against *M. tuberculosis* [24]. The SK-03-92 lead compound displayed MIC values lower than these other stilbenoid drugs. Preliminary work suggests the SK-03-92 drug may target a putative two-component system tied to late-stage competence in the bacteria [25].

Since *M. tuberculosis* survives and proliferates inside mammalian macrophages, new drugs that can kill the mycobacteria within macrophages are advantageous. Only a few antibiotics are able to penetrate through host cells to kill the intracellular bacteria [26,27]. SK-03-92 was able to reduce *M. tuberculosis* counts by more than two log_10_ values, even at a concentration of only 0.5 μg/mL. The *M. tuberculosis* viable count reduction caused by SK-03-92 surpassed the viable count loss shown for rifampin used at its highest concentration (1.6 log reduction). 

From this study, we have shown that the SK-03-92 lead compound has some efficacy against *M. tuberculosis* including single-drug-resistant mutants. Further, the ability to kill intracellular mycobacteria was an additional positive indication. Although SK-03-92 has displayed low toxicity in mice, several challenges remain. 

SK-03-92 is highly insoluble in aqueous solutions, and the concentration peaks at approximately 2 μg/mL in mouse plasma [28]. However, an analog of the SK-03-92 backbone that addressed the solubility issue and offered even greater efficacy could lead to the further development of this lead compound as an anti-mycobacterial drug.

## 4. Materials and Methods

### 4.1. Bacterial Strains, Growth Media, and Antibiotics

*Mycobacterium tuberculosis* strain H_37_Rv (ATCC 27294), isoniazid-resistant (INH^R^) *M. tuberculosis* strain SRI 1369, rifampin-resistant (RIF^R^) *M. tuberculosis* strain SRI 1367, and oxifloxacin-resistant (OXN^R^) *M. tuberculosis* strain SRI 4000 were used. All mycobacterial strains were grown on Middlebrook 7H9 agar plates or in Middlebrook 7H12 broth with oleic albumin dextrose catalase (OADC) enrichment (BD Biosciences, Sparks, MD, USA). The rifampin (RIF) and isoniazid (INH) antibiotics used were obtained from Sigma-Aldrich, St. Louis, MO, USA.

### 4.2. Minimum Inhibitory Concentration (MIC)

The broth microdilution assay was used according to the Clinical and Laboratory Standards Institute (CLSI) guidelines [29]. Briefly, testing was conducted in 96-well, U-bottom microtiter plates with an assay volume of 200 μL/well. Middlebrook 7H12 broth supplemented with OADC enrichment (BD Biosciences, Sparks, MD, USA) was added at a volume of 100 μL/well. The SK-03-92 in dimethyl sulfoxide (DMSO, ThermoFisher, Waltham, MA, USA) or RIF and INH in phosphate-buffered saline (PBS) were serially two-fold diluted in the same medium starting with a concentration of 50 μg/mL. Mycobacterial cells at a concentration of 1 × 10^6^ CFU/mL in Middlebrook 7H12 medium with OADC enrichment were added at an equal volume to each well, and the plates were incubated for 7 days at 37 °C with 90% humidity. The MIC values were reported as the lowest concentration of drug that inhibited growth of the bacteria, represented as μg/mL. Controls that were used included medium only, mycobacteria in medium, and rifampin or isoniazid as positive controls. Each strain was tested a minimum of two times with each drug.

### 4.3. Cytotoxicity Testing of SK-03-92

To determine the cytotoxicity of SK-03-92 at low (0.5 μg/mL), medium (5 μg/mL), and high concentrations (50 μg/mL), an MTT [3-(4,5-dimethylthiazol-2-yl)-2,5-diphenyltetrazolium bromide] assay (American Type Culture Collection, Manassas, VA, USA) was performed [30]. Mitomycin C (Sigma Aldrich, St. Louis, MO, USA) and DMSO were used as positive and negative controls, respectively. Rifampin was also assessed as a control at concentrations of 0.1, 1, and 10 μg/mL. The assay was repeated at least once in duplicate.

### 4.4. Intracellular Killing Assay

To assess inhibition of *M. tuberculosis* intracellularly by the SK-03-93 drug, the murine macrophage cell line J774A.1 (American Type Culture Collection, Manassas, VA, USA) was used. Briefly, J774A.1 macrophage cells were grown to confluence in tissue culture flasks using RPMI 1640 medium (Gibco, Waltham, MA, USA) containing 10% fetal bovine serum (FBS, Gibco) at 37 °C in the presence of 5% CO_2_. Macrophage cells were counted using a cell micrometer and diluted to approximately 2 × 10^5^ cells/mL, and 1 mL was pipetted into each well of a 12-well tissue culture plate. The cells were incubated in antibiotic-free RPMI 1640 medium with 10% FBS for 24 h. The media were replaced in the culture dishes and fresh medium was added to each well. Subsequently, 1 mL of *M. tuberculosis* strain H_37_Rv cells was added to each cell at a multiplicity of infection of 10:1. Dishes were incubated for 4 h to allow phagocytosis of the bacteria, and the cells washed twice with PBS to remove extracellular bacteria. Rifampin at concentrations of 0.01, 0.1, and 1.0 μg/mL was added in a 1 mL volume as a positive control, whereas PBS served as the untreated negative control. The SK-03-92 drug suspended in dimethyl sulfoxide to concentrations of 50 μg/mL, 5 μg/mL, and 0.5 μg/mL was added in a 1 mL volume to separate wells, and the plates were incubated for 7 days. At days 0 and 7, the assay medium was removed, and each well was washed 3 times with PBS. The macrophage cells were lysed with 1 mL of 0.25% sodium dodecyl sulfate in PBS. One hundred microliters of lysate from each well were removed and diluted in sterile PBS, and the lysate was plated on Middlebrook 7H10 agar plates (BD Biosciences). These plates were incubated at 37 °C for 16–21 days, and the colonies were enumerated. To determine the percent reduction in intracellular mycobacteria, the number of colonies from the lysed macrophage cells was divided by the number of colonies from the negative control wells and multiplied by 100. These assays were carried out in duplicate and repeated once. 

### 4.5. Statistics

A Student’s *t* test was used for statistical analyses. *p*-Values of ≤ 0.05 were considered significant.

## 5. Conclusions

The lead compound SK-03-92 displayed activity against several *M. tuberculosis* strains that included frontline-drug-resistant strains. Moreover, the SK-03-92 lead compound killed *M. tuberculosis* cells inside mammalian macrophages. To further develop SK-03-92 as an anti-mycobacterial drug, the solubility and peak plasma concentration level still need to be addressed by synthesizing analogs of the SK-03-92 backbone.

## Figures and Tables

**Figure 1 antibiotics-12-01385-f001:**
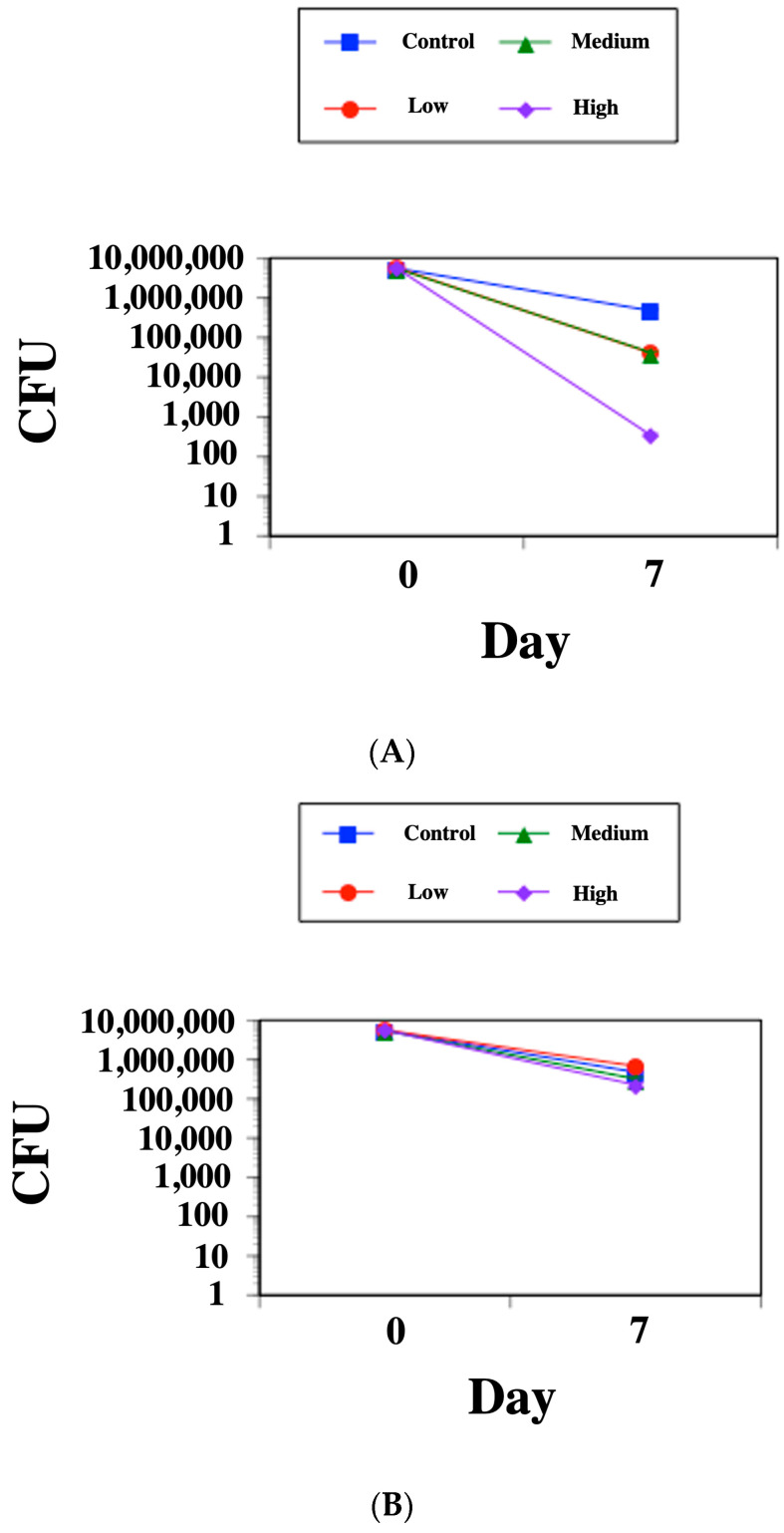
Macrophage intracellular killing by (**A**) SK-03-92 lead compound or (**B**) rifampin. Each drug that was tested was used at a low concentration (red circle), medium concentration (green triangle), or high concentration (purple diamond). An untreated PBS control (blue square) was used for each run. Assessments were carried out at day 0 and day 7 and were noted as colony forming units (CFUs). Each data point represents the mean ± standard deviation from two runs performed in duplicate.

**Table 1 antibiotics-12-01385-t001:** The SK-03-92 MIC results for four *M. tuberculosis* strains.

Strain	Drug Tested		
	SK-03-92	RIF ^a^	INH
*M. tuberculosis* H_37_Rv	6.25 ^b^	0.049	0.02
*M. tuberculosis* SRI 1369 (INH^R^) ^d^	6.25	0.039	NA ^c^
*M. tuberculosis* SRI 1367 (RIF^R^)	0.39	NA	0.02
*M. tuberculosis* SRI 4000 (OXN^R^)	0.78	0.78	NA

^a^ The drugs tested in the MIC analysis were RIF = rifampin, INH = isoniazid, and OXN = moxifloxacin. ^b^ The minimum inhibitory concentration (MIC) values are depicted as the mean values in μg/mL from three separate runs in microtiter plates. ^c^ NA = not assessed. ^d^ The superscript R denotes resistance to that antibiotic.

**Table 2 antibiotics-12-01385-t002:** Cytotoxicity analysis of SK-03-92 drug compared to rifampin on J774.A1 murine macrophage cell viability using an MTT assay.

Drug	% Viability	% Viability	% Viability
	(low conc.) ^a^	(med. conc.)	(high conc.)
SK-03-92	98 ± 2 ^b^	79 ± 3	<10
Rifampin	95 ± 1	93 ± 2	88 ± 2

^a^ For SK-03-92, the low concentration was 0.5 μg/mL, medium (med.) concentration was 5 μg/mL, and the high concentration was 50 μg/mL. The rifampin low concentration was 0.1 μg/mL, medium concentration was 1 μg/mL, and the high concentration was 10 μg/mL. ^b^ Represents the mean value from two runs performed in duplicate ± standard deviation.

## Data Availability

The data from this study are readily available from the author.

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
