# Peer review of "SK-03-92 Drug Kills Intracellular Mycobacterium tuberculosis"

_antibiotics, 2023, doi:10.3390/antibiotics12091385_

Round 1

Reviewer 1 Report

The manuscript "SK-03-92 drug kills intracellular Mycobacterium tuberculosis" discusses the latest results on lead compound SK-03-92. Although the compound is not new, the macrophage data is new and relevant considering the ongoing problem with TB globally. The manuscript is very well written, with close to no typographical errors. The presented data will appeal to a broader audience of this journal. The conclusions section is very brief, and I would like to know if the author could consider strengthening the conclusions by pointing out some limitations, as discussed on page 5, lines 125 - 132. Nonetheless, the manuscript is a good piece of work conducted by the group, and it will be interesting to see how far this compound progresses.

Author Response

I want to thank the reviewer for their constructive comments.  I have added limitation wording to the conclusion section as recommended by the reviewer.

Reviewer 2 Report

The article entitled ”SK-03-92 drug kills intracellular Mycobacterium tuberculosis” captures an interesting aspect of research on tuberculosis. The anti-Mtb activity of this compound was determined by the MICs in vitro and intracellular killing assay.  However, I recommend the following corrections.

1.      In Table 1, page 2: Author should be add more details of  a, b, c, and d.

2.      The results in Table 2, line 74-78. The high concentration of SK-03-92 had a high cytotoxicity to J774.A1 murine macrophage. Why did the author use this concentration to further study in the intracellular killing assay?

Author Response

I want to thank the reviewer for their constructive comments.  For each of the concerns that were raised, my comments are noted below.

  1. Greater detailed information has been added to the footnotes for Table 1.
  2. The SK-03-92 lead compound was sent to the Southern Research Institute for their tuberculosis screening. To test for cytotoxicity and intracellular killing, their protocol is to test a medium range concentration around the MIC for the drug, a ten-fold lower concentration for the low concentration, and a ten-fold higher concentration for the high concentration.  Thus, the high concentration set by their protocol was 50 mg/mL.

Reviewer 3 Report

The authors have conducted experiments to assess the impact of SK-03-92 on various strains of Mycobacterium tuberculosis (Mtb). While the authors claims that SK-03-92 could potentially serve as an effective agent against Mtb, I find that this study is still in its preliminary stages, and I believe additional evidence is necessary to substantiate the presented conclusions.

A significant consideration is the notably higher minimum inhibitory concentration (MIC) of SK-03-92 compared to existing drugs such as RIF or INH. Additionally, the drug's cytotoxicity is considerable. It raises questions as to why the experiments were conducted using concentrations that led to the demise of approximately 20% of host cells at medium drug doses and 90% at high drug doses. Under such cytotoxic conditions, host cells are subject to substantial stress, making them less than ideal for this type of experimentation.

 It remains unclear whether this compound merely joins the ranks of numerous antibacterial agents or if it specifically targets tuberculosis. The authors have not provided any data demonstrating the drug's specificity to tuberculosis.

Minor points- The Mtb strain used and untreated CFU data are missing in killing experiments. Addressing these concerns and providing more data would enhance the study's reliability.

The quality of english is not as par as it should be for scientific papers. 

Author Response

I want to thank the reviewer for their constructive comments.  The purpose of this study was to get a preliminary assessment of the efficacy of SK-03-92 against M. tuberculosis.  We agree with the reviewer that the work is only preliminary.  Thus, it was submitted as a short communication rather than a full research article.  With that said, SK-03-92 had an okay MIC value against regular M. tuberculosis (MIC = 6.25 mg/mL).  Against single drug resistant strains, SK-03-92 displayed MIC values between 0.39 and 6.25 mg/mL.  The 0.39 mg/mL MIC number noted for the rifampin-resistant strain is in line with a good MIC susceptibility result.  The reviewer is correct that a high concentration of SK-03-92 drug severely damaged the macrophage cells.  Through the NIH contract to screen potential anti-M. tuberculosis compounds, they set the high concentration to 50 mg/mL.  However, when the low concentration of SK-03-92 with minimal cytotoxicity was used in the intracellular killing assay, the mean log drop in the viable count was higher than all three concentrations of rifampin that were used.  We have shown that SK-03-92 has efficacy only against Gram-positive bacteria as well as mycobacteria.  We did not state that the drug is mycobacteria specific.  To address your minor points, the M. tuberculosis strain that was used in the intracellular killing assay has been added to the text.  In the manuscript, we have also stated that the PBS control is the untreated control in the killing experiments.  We are aware that additional analogs need to be synthesized and tested.

Round 2

Reviewer 3 Report

I hold a differing viewpoint from the author's response regarding submission to short communications and not full research article. In my opinion, disregarding the establishment of a well-structured experiment that yields conclusive evidence, rather than speculative outcomes, is not advisable.  

Employing lethal concentrations of drugs that result in the death of host cells is an unsuitable experimental approach for drawing conclusions about the intracellular killing if bacteria.